# Emerging Antibiotic Resistance Patterns Affect Visual Outcome Treating Acute Endophthalmitis

**DOI:** 10.3390/antibiotics11070843

**Published:** 2022-06-23

**Authors:** Xia-Ni Wu, Yi-Hsing Chen, Lazha Sharief, Ahmed Al-Janabi, Nura Al Qassimi, Sue Lightman, Oren Tomkins-Netzer

**Affiliations:** 1NIHR Biomedical Research Centre, Moorfields Eye Hospital, London EC1V 2PD, UK; xianiwu@gmail.com (X.-N.W.); yi-hsing.chen.14@alumni.ucl.ac.uk (Y.-H.C.); lazha_talat@yahoo.com (L.S.); atj_83@yahoo.com (A.A.-J.); n.alqassimi@gmail.com (N.A.Q.); s.lightman@ucl.ac.uk (S.L.); 2Institute of Ophthalmology, University College London, London EC1V 9EL, UK; 3Department of Ophthalmology, Chang Gung Memorial Hospital, Taoyuan 10507, Taiwan; 4Department of Ophthalmology, Lady Davis Carmel Medical Center, Ruth and Bruch Rappaport Faculty of Medicine, Technion Israel Institute of Technology, Haifa 3200003, Israel

**Keywords:** endophthalmitis, antibiotic resistance, corticosteroids, vancomycin, moxifloxacin

## Abstract

Background: Examining the effect of antibiotic resistance, use of intravitreal antibiotics and systemic corticosteroids on visual outcome of eyes with acute endophthalmitis. Methods: We included 226 eyes with acute endophthalmitis, treated using a standardized protocol. Visual outcome up to 12 months was assessed related to biopsy results, antibiotics resistance and treatment regimens. Results: Vitreous biopsies were more likely to be culture-positive (41.1%) than anterior chamber biopsies (21.6%, *p* < 0.0001). Antibiotic resistance for amikacin was found in 19 eyes (24.7%), vancomycin in 29 eyes (31.5%) and moxiflocacin in 14 eyes (16.1%). At presentation 91.53% of eyes had BCVA < 20/40, reducing by 1 month to 69.94% (*p* < 0.0001) and remaining stable at 12 months. There was no difference in visual outcome for those receiving early systemic corticosteroids. Endophthalmitis following cataract surgery (OR 1.66, 1.04–2.66 95% CI, *p* = 0.03) and receiving intravitreal vancomycin (OR 3.15, 1.18–8.42 95% CI, *p* = 0.02) were associated with a greater chance of final BCVA ≥ 20/40. Conclusion: Using vitreous taps with intravitreal antibiotics, despite an increase in resistance to both vancomycin and moxifloxacin, results in a final BCVA > 20/200 in half of eyes and ≥20/40 in a third. Early treatment with intravitreal antibiotics should not be delayed.

## 1. Introduction

Endophthalmitis is a sight-threatening disease and has a high risk of poor visual outcome and even complete vision loss [1]. The profile of endophthalmitis has changed over the past 30 years. In post-cataract surgery endophthalmitis the incidence has been decreasing significantly, ranging from 0.13% to 0.7% in the 1990s [1,2] to 0.04% in 2013–2017 in the United States [3]. Additionally, Gram-negative bacteria play a more important role in the bacterial profile isolated from cases of endophthalmitis [4], resulting in poorer visual outcome in many cases [5].

Patients presenting with endophthalmitis require immediate ocular sampling and administration of broad-spectrum intravitreal antibiotics. Choice of antibiotics used, changing antibiotic resistance patterns and ocular sampling techniques affect the chance of successful diagnosis and control of the infection. Studies report reduced susceptibility to ciprofloxacin [6], vancomycin [7,8] and recently to moxifloxacin [9,10]. The Endophthalmitis Vitrectomy Study (EVS) reported that there is no additional benefit in the use of systemic antibiotics [1]; however, oral antibiotics with better blood retinal barrier penetration including moxifloxacin still have a beneficial outcome to the clinical course of infection and visual outcome [11,12], with moxifloxacin replacing ciprofloxacin [11]. Furthermore, the early use of systemic corticosteroids may also influence the clinical outcome, ocular complications, and visual outcome of these patients [13].

We now report the pathogenic organisms, their antibiotic susceptibility, and clinical outcomes following treatment of endophthalmitis using a standardized clinical protocol including vancomycin, adjunctive systemic moxifloxacin, and systemic corticosteroids. We examine the effect of these factors and treatment approaches on visual outcome.

## 2. Results

The study included 233 episodes of endophthalmitis in 232 eyes (122 right) of 232 pts (92 female). The median age at presentation was 66.76 years (IQR 48.13–76.06 years). Seven cases were diagnosed as caused by fungal pathogens and were excluded from further analysis. Of the remainder 226 cases the most common mechanisms were following cataract surgery (*n* = 71, 31.42%), filtration surgery (*n* = 63, 27.88%) or intravitreal injections (*n* = 39, 17.26%). Other less common causes were following trauma, post-vitrectomy, or endogenous endophthalmitis. The median time from diagnosis to biopsy was 0 days (IQR 0.0–0.0 days) and the length of follow-up was 43.15 ± 2.65 months (817.92 eye-years).

### 2.1. Antibiotic Resistance

Information on ocular sampling was collected for 221 eyes (97.8%). Anterior chamber (AC) biopsies were performed in 162 eyes (73.3%) and vitreous biopsies were performed in 219 eyes (99.1%). Ocular samples were culture-positive in 99 cases (44.8%) with vitreous biopsies positive in 90 eyes (41.1%) and AC biopsies in 35 (21.6%) eyes (*p* < 0.0001). Among eyes with a culture-positive vitreous biopsy, 25 (27.8%) also had a positive AC biopsy, and in 24 of those eyes, the identified organism was the same. In one case, following cataract surgery, the vitreous biopsy identified *Corynebacterium tuberculostearicum* in the vitreous biopsy and *Micrococcus luteus* in the AC. There were nine eyes (25.7%) with positive cultures from AC biopsies but a negative vitreous biopsy. Gram staining identified gram-positive bacteria in 64 cases (64.7%). The most common pathogen isolated was *Staphylococcus epidermidis*, isolated in 24.2% (*n* = 24) of positive cultures, followed by *Streptococcus mitis* in 11.1% (*n* = 11), *Haemophilus influenza* in 8.1% (*n* = 8) and *Pseudomonas aerugenosa* in 5.1% (*n* = 5, Appendix A). Antibiotic resistance patterns were tested for all culture-positive biopsies and information was available for 77 eyes (77.78%). Resistance included amikacin in 19 eyes (24.7%), vancomycin in 29 eyes (31.5%) and moxiflocacin in 14 eyes (16.1%). Among gram-positive bacteria, vancomycin resistance was found in five cases (7.8%). There were three eyes with resistance to all three drugs: one bleb-related endophthalmitis caused by *Streptococcus mitis-oralis*; one bleb-related endophthalmitis caused by *Haemophilus influenza* and one endogenous endophthalmitis caused by *Aeromonas sobras*.

### 2.2. Early Treatment for Endophthalmitis

At presentation intravitreal injections of vancomycin were administered in 185 eyes (93.43%), amikacin in 179 eyes (91.32%) and ceftazidime in eyes eyes (1.01%). Intravitreal steroids were injected in 35 eyes (15.63%), the majority in bleb-related endophthalmitis (*n* = 33). During the first 24 h 36 eyes (15.92%) were treated using systemic corticosteroids at a dose of 1 mg/kg. Systemic antibiotics were used to treat 182 eyes (80.53%): 157 (86.74%) were given systemic moxiflocacin, 15 (6.64%) amoxicillin/clavulanic acid and 10 (4.42%) ciprofloxacin. Non-protocol antibiotic regimens were used at the physician’s discretion, based on patients’ known allergies. Early vitrectomy, during the first 24 h, was performed in nine eyes (3.98%). Systemic corticosteroids were started after the first 24 h in 128 eyes (74.85%). Repeat intravitreal biopsies were performed in 23 eyes (10.18%) at a median time of 2.0 days (IQR 2–5.25 days).

Average BCVA at presentation was 1.82 ± 0.06 LogMAR improving by 1 month to 1.27 ± 0.0.8 LogMAR and remaining stable by 12 months and final follow-up (Figure 1). At presentation, 91.53% of eyes had vision loss (BCVA < 20/40), of which 79.64% had severe vision loss (SVL, ≤20/200). By 1 month the percent of eyes with vision loss reduced to 69.94%, of which 49.13% had SVL (*p* < 0.0001), remaining stable by 12 months (Figure 2). At presentation, there were 27 eyes with BCVA of PL, of which one underwent early pars plana vitrectomy (<24 h). At one month follow-up BCVA improved to better than PL in 60.9% (*n* = 14) of eyes and remained stable at 57.1% (*n* = 12) by 12 months. By 12 months average BCVA was 2.19 ± 0.19 LogMAR, 67.7% had SVL and 20.4% had BCVA ≥ 20/40.

Visual outcome at 1 month was better for eyes with culture-negative than culture-positive biopsies (1.64 ± 0.09 LogMAR vs. 2.06 ± 0.08 LogMAR, *p* = 0.001). This difference remained significant throughout follow-up and by 12 months culture-negative eyes had a BCVA of 1.02 ± 0.11 LogMAR compared with 1.74 ± 0.14 LogMAR for culture-positive eyes (*p* < 0.0001). Drug resistance to amikacin resulted in less improvement in visual acuity at 6 (*p* = 0.003) and 12 months (*p* = 0.002). No such effect was found for drug resistance to vancomycin or moxiflocacin. There was no difference in visual outcome among eyes that received systemic corticosteroids during the first 24 h (*n* = 36) and those that did not. Similarly, there was no difference between eyes that underwent a vitrectomy during the first 24 h (*n* = 9) and those treated using a tap-and-inject approach.

The three main etiologies were post-cataract surgery, post-filtration surgery, and post-intravitreal injection. Table 1 summarizes a comparison of clinical findings and outcome measures of these groups. There was a significant difference in age at presentation between groups, driven by a younger age of patients in the post-filtration surgery group. Eyes in the post-filtration group with positive cultures were more likely to have gram-negative species (*p* = 0.009), with a greater likelihood of amikacin resistance compared to eye post-cataract surgery (*p* = 0.02) or post intravitreal injection eyes (*p* = 0.014). BCVA improved during the first 12 months for eyes in all groups, though significantly less for eyes in the post-filtration group (Figure 3). By final follow-up, 48.5% of eyes in the post-cataract surgery group had a final BCVA ≥ 20/40 (no vision loss) and 30.9% had SVL.

We examined factors associated with a final visual acuity of no vision loss (BCVA ≥ 20/40) and found on multivariate analysis that endophthalmitis following cataract surgery (OR 1.66, 1.04–2.66 95% CI, *p* = 0.03) and receiving treatment with intravitreal vancomycin (OR 3.15, 1.18–8.42 95% CI, *p* = 0.02) were associated with a greater chance of final BCVA ≥ 20/40 (Table 2). Interestingly, a negative culture was significant on univariate analysis (OR 1.6, 1.02–2.49, *p* = 0.04) but lost significance when other factors were accounted for. Resistance to one antibiotic or more did not affect the risk of vision loss.

## 3. Discussion

The global trend of increasing antibiotic resistance has significant implications for medical care, morbidity, and mortality. Endophthalmitis is managed with empirical intravitreal antibiotics, typically vancomycin for gram-positive coverage and ceftazidime or amikacin for gram-negative organisms [14]. In this study, we report significant resistance to amikacin, moxifloxacin, and vancomycin. Vancomycin resistance was found in a third of cultures and specifically in gram-positive endophthalmitis. Recent studies identified increased rates of vancomycin resistance in culture-positive endophthalmitis, most importantly among gram-positive bacteria—accounting for up to 11% of isolates [8,15,16]. Despite documented resistance in cultures, we found that drug resistance was not associated with a risk of vision loss and that the use of intravitreal vancomycin was significantly associated with a greater chance of final BCVA ≥ 20/40. Intraocular vancomycin concentrations following intravitreal injection are much higher than the MIC [17], so that clinical response may be incongruent to in-vitro culture resistances. Nevertheless, increasing resistance rates warrant continued monitoring in order to identify when empirical use of vancomycin may become inadequate.

Systemic antibiotic therapy in exogenous endophthalmitis is widely used, though evidence from clinical studies is contradictory [1,11]. Oral moxifloxacin has good penetration across the blood-ocular barrier and achieves therapeutic intra-ocular levels above the MIC of many gram-positive and gram-negative bacteria [18,19,20]. In our study use of oral moxifloxacin did not correlate with better visual outcome. Its role in the treatment of endophthalmitis is not supported by this study as its impact is outweighed by the effect of intravitreal antibiotics. Expedient intravitreal administration of broad-spectrum antibiotics remains the main objective of early management. Increasing rates of resistance have been reported [21,22,23], and in this study, we showed 16.7% resistance across all isolates. Fourth-generation fluoroquinolones, such as delafloxacin, demonstrate promising MIC effects in in-vitro studies and may be considered as adjunctive treatment in the future [24].

Some ophthalmologists use early systemic corticosteroids [1,13], with others concerned with inducing an immunosuppressive state. A recent retrospective study of 133 eyes with endophthalmitis found that the use of oral steroids was associated with greater improvement in visual acuity and the likelihood of three lines improvement [13]. Our results support that use of early corticosteroids was safe, but did not affect the final visual acuity or reduce the chance of eyes achieving a BCVA ≥ 20/40. Early use of corticosteroids could be considered in cases with overt inflammation at presentation to expedite response.

Visual acuity improved in our cohort, including among eyes with PL at presentation. The EVS examined visual outcome among eyes with endophthalmitis post-cataract surgery and demonstrated that by 12 months 74% achieved BCVA > 20/200 and 53% ≥20/40 [1]. Our results for the subgroup of endophthalmitis post-cataract surgery demonstrated that 69.1% achieved a BCVA > 20/200 and 48.5% ≥20/40. Our results were also comparable among eyes presenting with PL, despite most not undergoing primary vitrectomy. This reflects the effect of early management protocols, the use of broad-spectrum intravitreal antibiotics, and the administration of concomitant anti-inflammatory measures [25,26,27]. Final BCVA for eyes with additional pathologies, such as AMD or glaucoma may be worse, reflecting damage to visual acuity from the underlying condition [28].

Strengths of this study include a large dataset treated according to an acute management protocol, consistency in data collection, and samples that were sent to a single laboratory with standardized testing and reporting. The retrospective design has inherent limitations, which include incomplete data, selection, and follow-up biases. However, the large cohort and extensive follow-up allowed us to examine different treatment approaches and identify their effect on lasting visual outcome. Statistically meaningful analyses were limited by the small sample sizes in some subgroups such as vitrectomy performed within 24 h. Nevertheless, using a large cohort with cases related to different mechanisms, we were able to explore their effect on outcome as well as that on antibiotic resistance patterns and treatment choices. Comparison to cohorts of eyes that underwent early vitrectomy may provide useful data regarding its role in the management of endophthalmitis.

In this study, we found that vitreous taps were more reliable than aqueous sampling, with an increasing laboratory resistance to both vancomycin and moxifloxacin. Almost half of eyes achieved a final BCVA better than 20/200 and a third ≥20/40, despite most undergoing a tap-and-inject approach. Laboratory resistance to antibiotics did not result in increased inflammation or a poorer visual acuity outcome. Protective factors against vision loss were antecedent cataract surgery and use of intravitreal vancomycin. Despite an increase in drug resistance, early intravitreal injection of broad spectrum antibiotics remains an effective treatment for treating acute endophthalmitis, is the most important action to consider and should not be deferred. Future studies must continue to monitor changes in resistance patterns and the effectiveness of the current treatment approach, in order to maintain clinical outcome in these cases.

## 4. Materials and Methods

### 4.1. Study Population

This retrospective, longitudinal study examined patients diagnosed with acute endophthalmitis, seen at Moorfields Eye Hospital (UK) between 2006 and 2017. The study received institutional ethics approval (ethics approval ROAD16039) and adhered to the tenets of the Helsinki declaration. Patients were included in the study if they presented with reduced vision, an injected painful eye, upon examination were found to have intraocular inflammation involving the anterior chamber and vitreous cavity and diagnosed with endophthalmitis. Following clinical diagnosis, eyes underwent immediate vitreous biopsy by pars plana vitrectomy or vitreous tap, anterior chamber biopsy, and intravitreal injection of vancomycin 2 mg/0.1 mL and either amikacin 0.4 mg/0.1 mL or ceftazidime 2.25 mg/0.1 mL. Systemic moxifloxacin 400 mg od was given for ten days and in some cases, systemic corticosteroids were added. Eyes were reviewed at 48 h and repeat biopsy was performed in cases with no clinical improvement. Vitreous biopsies were Gram-stained and cultured for 72 h. Cultivated organisms and antibiotic susceptibilities were recorded.

A review of patients’ clinical and pathology data was performed and information was gathered regarding patient demographics, any known ocular procedure, systemic condition or trauma occurring within six months prior to the diagnosis of acute endophthalmitis, microbiological culture results and antibiotic susceptibilities, local and systemic treatments, ocular complications and best-corrected visual acuity (BCVA) at baseline and at months 1, 3, 6, 12, and final follow-up. Visual acuities were converted to LogMAR values for statistical analysis. Additionally, count fingers, hand movements, perception of light (PL), and no perception of light vision were converted to 2.0, 2.3, 2.6, and 2.9, respectively [29]. Moderate vision loss (MVL) was defined as BCVA < 20/40 and >20/200. Severe vision loss (SVL) was defined as BCVA ≤ 20/200.

### 4.2. Statistical Analysis

Analyses were performed by using SPSS statistical software version 22 (IBM, Chicago, IL, USA). Continuous data are presented as mean ± standard error of the mean (SEM) while categorical data are presented as median and interquartile range (IQR). Multivariate analysis and relative risks for vision loss were calculated using a Cox regression model. Culture rates and vision loss patterns were compared using a Pearson’s Chi-square test. Comparisons between etiology groups were calculated using the Kruskal–Wallis test. The Dunn-Bonferroni post hoc test was used to evaluate pairwise comparisons. Differences in BCVA between groups were calculated using one-way ANOVA. A *p*-value of <0.05 was considered to be significant.

## Figures and Tables

**Figure 1 antibiotics-11-00843-f001:**
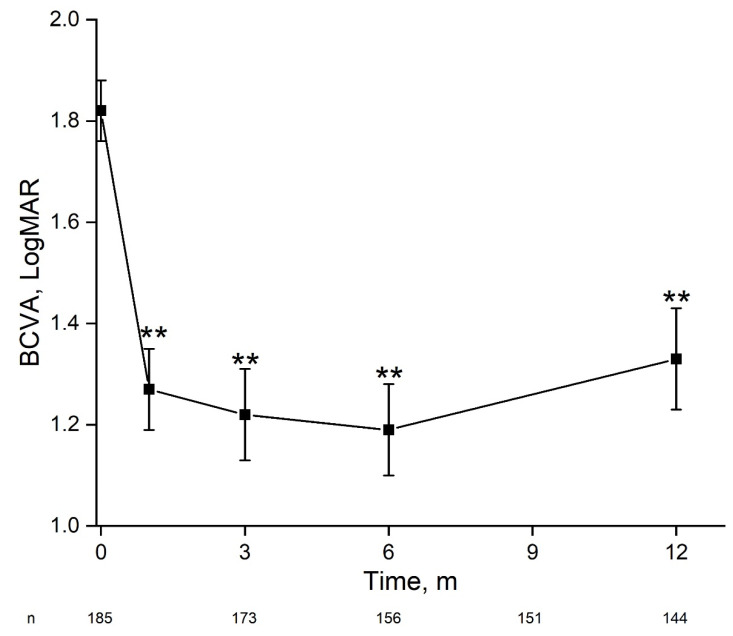
Change in visual acuity during 12 months following acute endophthalmitis. For the entire cohort, there was a significant and stable improvement in best-corrected visual acuity within one month. BCVA—best-corrected visual acuity, m—months. **—*p* < 0.01 (one-way ANOVA).

**Figure 2 antibiotics-11-00843-f002:**
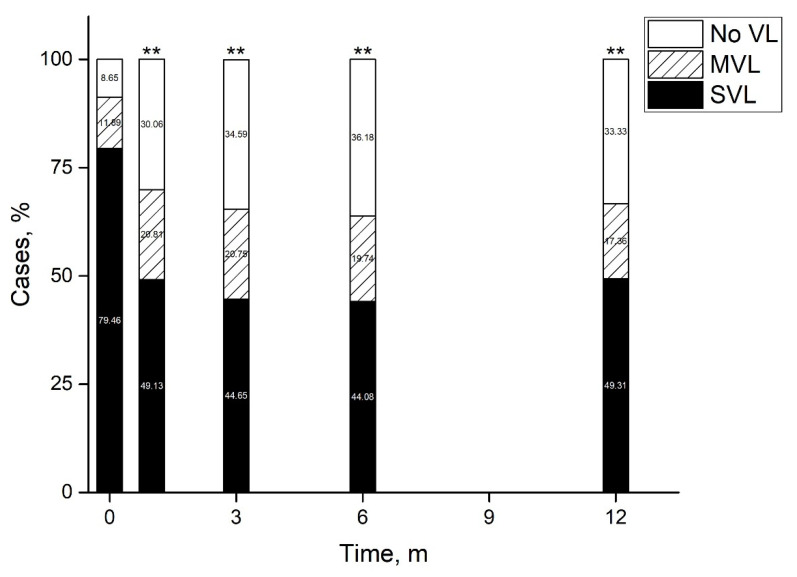
Vision loss patterns following acute endophthalmitis. By 12 months half of eyes had improved to a visual acuity better than 20/200 and a third had no vision loss. **—*p* < 0.01 (Pearson’s Chi-square test). m—months, MVL—moderate vision loss, SVL—severe vision loss, VL—vision loss.

**Figure 3 antibiotics-11-00843-f003:**
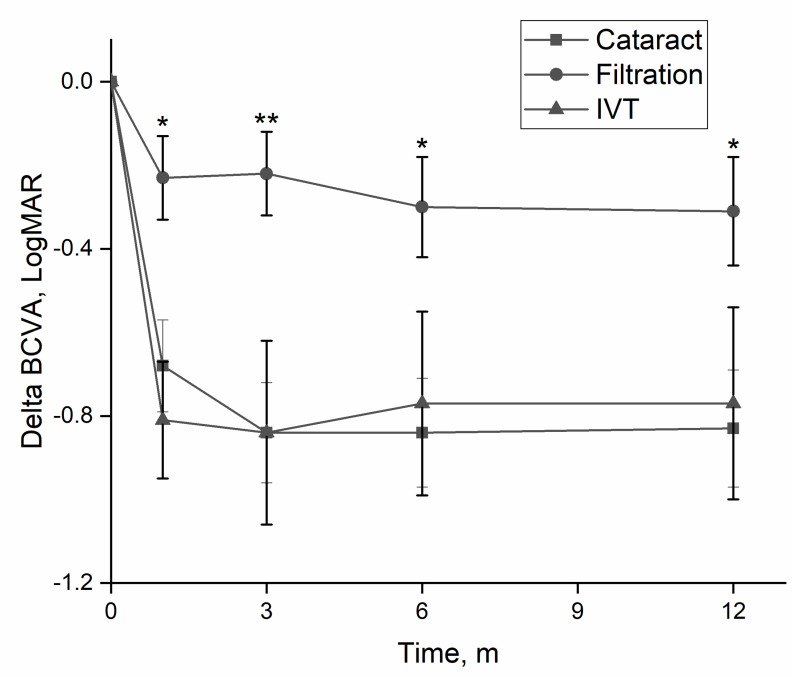
Change in visual acuity according to etiology. Eyes with endophthalmitis following filtration surgery had significantly less improvement in visual acuity, than eyes following cataract surgery or intravitreal injections. BCVA—best-corrected visual acuity, m—months. *—*p* < 0.05; **—*p* < 0.01 (one-way ANOVA).

**Table 1 antibiotics-11-00843-t001:** Clinical findings and visual outcome among eyes with endophthalmitis related to post-cataract surgery, post-filtration surgery, and post intravitreal injections.

	Post-Cataract Surgery, *n* (%)	Post-Filtration Surgery, *n* (%)	Post-Intravitreal Injection, *n* (%)	*p*-Value
Eyes, *n*	71	63	39	
Age at presentation, yrs, mean ± SEM	69.5 ± 1.5	61.4 ± 2.2	71.5 ± 2.5	0.004
Follow-up, m, mean ± SEM	47.4 ± 4.9	46.48 ± 4.3	34.4 ± 5.3	0.13
Culture-positive	29 (40.9)	25 (39.7)	17 (43.6)	0.42
Moxifloxacin resistance	4 (16.7)	5 (25.0)	1 (5.6)	0.27
Vancomycin resistance	7 (26.9)	8 (38.1)	2 (10.5)	0.14
Amikacin resistance	3 (14.0)	9 (52.94)	2 (11.1)	0.007
Final BCVA ≥20/40	33 (48.5)	21 (34.4)	9 (25)	0.06
Final BCVA >20/200	47 (69.1)	30 (49.2)	18 (50)	0.18

BCVA—best-corrected visual acuity.

**Table 2 antibiotics-11-00843-t002:** Factors related to vision loss.

	Crude OR (95% CI)	*p*-Value	Refined OR (95% CI)	*p*-Value
Risk factor				
Age	1.01 (0.99–1.02)	0.23		
Cataract Sx	1.4 (0.94–2.07)	0.1	1.66 (1.04–2.66)	0.03
Post Intravitreal inj	1.14 (0.67–1.94)	0.64		
Culture Negative	1.6 (1.02–2.49)	0.04	1.67 (0.98–2.84)	0.06
Moxifloxacin sensitivity	1.38 (0.5–3.76)	0.53		
Vancomycin sensitivity	1.04 (0.42–2.55)	0.93		
Amikacin sensitivity	0.92 (0.34–2.49)	0.86		
IVT Vancomycin	1.86 (0.88–3.93)	0.1	3.15 (1.18–8.42)	0.02
IVT Amikacin	1.44 (0.71–2.9)	0.31		
PPV	1.01 (0.62–1.65)	0.97		
Early Prednisolone	0.622 (0.35–1.12)	0.11		
Oral Moxifloxacin	0.91 (0.48–1.72)	0.76		
IVT Corticosteroids	0.95 (0.54–1.68)	0.87		
Need for repeat Inj	0.61 (0.35–1.07)	0.08	0.56 (0.29–1.11)	0.09

IVT—intravitreal.

## Data Availability

The data presented in this study are available on request from the corresponding author. The data are not publicly available due to patient privacy.

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
