# Peer review of "Emerging Antibiotic Resistance Patterns Affect Visual Outcome Treating Acute Endophthalmitis"

_antibiotics, 2022, doi:10.3390/antibiotics11070843_

Round 1

Reviewer 1 Report

Recommendation : Accept

This article is very well written, novel and interesting. I think this manuscript must be published. However, I have few minor comments and suggestions for authors as given below: 

In conclusion or discussion section, please add some future plans in this scope of study, regarding its applicability and benefits and how do you see to involve these suggested approach and methods.

Author Response

Thank you for the review and comments of our manuscript (#antibiotics-1720081). We have addressed the reviewers’ comments individually and listed specific changes made to the manuscript in the form of a table as suggested.

Reviewer Comment

Response

Change

Reviewer 1

This article is very well written, novel and interesting. I think this manuscript must be published. However, I have few minor comments and suggestions for authors as given below: 

In conclusion or discussion section, please add some future plans in this scope of study, regarding its applicability and benefits and how do you see to involve these suggested approach and methods.

Thank you for your comment. Identifying resistance patterns and clinical response to treatment is extremely important for the management of acute endophthalmitis. Our results support the early use of vitreous taps and intravitreal antibiotics, which remain effective despite increases in drug resistance. Continual monitoring of resistance to antibiotics is needed to ascertain that the treatment we provide remain effective. We add text to the discussion.

Page 18 Line 9, " Despite an increase in drug resistance, early intravitreal injection of broad spectrum antibiotics remains an effective treatment for treating acute endophthalmitis, is the most important action to consider and should not be deferred. Future studies must continue to monitor changes in resistance patterns and the effectiveness of the current treatment approach, in order to maintain clinical outcome in these cases."

We corrected any grammar or spelling mistakes that were found.

We hope this revised manuscript is acceptable for publication in your journal.

Sincerely,

Oren Tomkins-Netzer MD, PhD

Reviewer 2 Report

In this paper, the authors treated using a standardized protocol with acute endophthalmitis to make sure about the effect of antibiotic resistance. They concluded that early treatment with intravitreal antibiotics should not be delayed. Some significant concerns (see below) deter the enthusiasm.

1. The paper title is “antibiotic resistance patterns affect visual outcome treating acute endophthalmitis”, so the important thing is to make sure about the antibiotic resistance and visual outcome, but the use of antibiotics and treatment of each patient is different in this study, the research needs to be more refined and rigorous.
2. Antibiotic resistance is not a novel topic, but it has great significance in the clinical treatment. The introduction is insufficient. It should describe the progress of antibiotic resistance, the characteristics of this study, the differences and significance.
3. The author made a comparison of endophthalmitis patients with three main etiologies, which has some significance in prognostic. But the patients' etiologies in clinical settings are not controlled by doctors. Therefore, if the author classified patients with antibiotic resistance, studying the treatment effect from this direction, not etiologies, will be more meaningful.

Author Response

Thank you for the review and comments of our manuscript (#antibiotics-1720081). We have addressed the reviewers’ comments individually and listed specific changes made to the manuscript in the form of a table as suggested.

Reviewer Comment

Response

Change

Reviewer 2

1. The paper title is “antibiotic resistance patterns affect visual outcome treating acute endophthalmitis”, so the important thing is to make sure about the antibiotic resistance and visual outcome, but the use of antibiotics and treatment of each patient is different in this study, the research needs to be more refined and rigorous.

We thank the reviewer for his comment. Patients with acute endophthalmitis are treated as an acute event and require immediate diagnosis and treatment. Current approaches to these patients included immediate ocular sampling and intravitreal antibiotics. However there are variations in the methods of sampling and drugs used.

In this study we analysed the effect of the different treatment approaches (antibiotics used, use of corticosteroids, sampling technique), as well as that of current antibiotics resistance patterns. We examined the effect of each and in combination on the visual outcome of patients. We clarify this in the text.

Page 4 Line 9, " Patients presenting with endophthalmitis require immediate ocular sampling and administration of broad-spectrum intravitreal antibiotics. Choice of antibiotics used, changing antibiotic resistance patterns and ocular sampling technique affect the chance of successful diagnosis and control of the infection."

Page 5 Line 2, "We now report the pathogenic organisms, their antibiotic susceptibility and clinical outcomes following treatment of endophthalmitis after the use of a standardised clinical protocol including vancomycin, adjunctive systemic moxifloxacin and systemic corticosteroids. We examine the effect of these factors and treatment approaches on visual outcome."

Page 10 Line 11, "Drug resistance to amikacin resulted in less improvement in visual acuity at 6 (p=0.003) and 12 months (p=0.002). No such effect was found for drug resistance to vancomycin or moxiflocacin."

Page 13 Line 11, "Resistance to one antibiotic or more did not affect the risk of vision loss."

2. Antibiotic resistance is not a novel topic, but it has great significance in the clinical treatment. The introduction is insufficient. It should describe the progress of antibiotic resistance, the characteristics of this study, the differences and significance.

Page 4 Line 9, " Patients presenting with endophthalmitis require immediate ocular sampling and administration of broad-spectrum intravitreal antibiotics. Choice of antibiotics used, changing antibiotic resistance patterns and ocular sampling technique affect the chance of successful diagnosis and control of the infection. Studies report reduced susceptibility to ciprofloxacin,6 vancomycin7,8 and recently to moxifloxacin.9,10 The Endophthalmitis Vitrectomy Study (EVS) reported that there is no additional benefit in the use of systemic antibiotics;1 however, oral antibiotics with better blood retinal barrier penetration including moxifloxacin still have a beneficial outcome to the clinical course of infection and visual outcome,11,12 with moxifloxacin replacing ciprofloxacin.11"

Page 23 Line 4, "6. Asbell PA, Sanfilippo CM, Mah FS. Antibiotic susceptibility of bacterial pathogens isolated from the aqueous and vitreous humour in the Antibiotic Resistance Monitoring in Ocular micRoorganisms (ARMOR) Surveillance Study: 2009-2020 update. J Glob Antimicrob Resist 2022;23:236-240.

7. Peck TJ, Patel SN, Ho AC. Endophthalmitis after cataract surgery: an update on recent advances. Curr Opin Ophthalmol 2021;32:62-68.

8. Relhan N, Albini TA, Pathengay A, et al. Endophthalmitis caused by Gram-positive organisms with reduced vancomycin susceptibility: literature review and options for treatment. Br J Ophthalmol 2016;100:446-52.

9. Chang VS, Schwartz SG, Davis JL, Flynn HW. Endophthalmitis following cataract surgery and intracameral antibiotic: Moxifloxacin resistant Staphylococcus epidermidis. Am J Ophthalmol Case Rep .2018;8:127-130.

10. Kato JM, Tanaka T, de Oliveira LMS. Surveillance of post-cataract endophthalmitis at a tertiary referral center: a 10-year critical evaluation. Int J Retina Vitreous 2021;16:14."

3. The author made a comparison of endophthalmitis patients with three main etiologies, which has some significance in prognostic. But the patients' etiologies in clinical settings are not controlled by doctors. Therefore, if the author classified patients with antibiotic resistance, studying the treatment effect from this direction, not etiologies, will be more meaningful.

We agree with the reviewer that there are various effectors on treatment outcome, including drug resistance and treatment choice. In our study we found that drug resistance to amikacin was associated with significantly less visual acuity improvement at 6 and 12 month follow-up. There was no such effect regarding resistance to vancomycin or moxifloxacin. There was no effect for resistance to any drug on risk of vision loss. We now clarify this in the results and discussion.

Page 10 Line 11, " Drug resistance to amikacin resulted in less improvement in visual acuity at 6 (p=0.003) and 12 months (p=0.002). No such affect was noted with regards to drug resistance to vancomycin or moxiflocacin."

Page 14 Line 29, " In this study we report significant resistance to amikacin, moxifloxacin and vancomycin. Vancomycin resistance was found in a third of cultures and specifically in gram-positive endophthalmitis. Recent studies identified increasing rates of vancomycin-resistance in culture-positive endophthalmitis, most importantly among gram-positive bacteria- accounting up to 11% of isolates.15-17 Despite documented resistance in cultures we found that drug resistance was not associated with a risk of vision loss and that the use of intravitreal vancomycin was significantly associated with a greater chance of final BCVA ≥20/40."

We corrected any grammar or spelling mistakes that were found.

We hope this revised manuscript is acceptable for publication in your journal.

Sincerely,

Oren Tomkins-Netzer MD, PhD

Reviewer 3 Report

Wu and colleagues report a very interesting study in which they examined the effect of antibiotic resistance and the use of intravitreal antibiotics and systemic corticosteroids on visual outcome in patients with acute endophthalmitis. Increasing antibiotic resistance constitutes a considerable therapeutic challenge in the clinical setting of acute endophthalmitis. The authors studied 226 eyes treated using a standardized protocol. They assessed the visual outcome up to 12 months related to biopsy results, antibiotics resistance, and treatment regimens. They concluded that the use of vitreous taps with intravitreal antibiotics results in satisfactory improvement in BCVA, despite an increase in resistance to both vancomycin and moxifloxacin. Thus, they recommend that intravitreal antibiotics should be administered as early as possible. On the other hand, steroids – although safe – have no effect on the final visual outcome.

The study is well designed and methodologically sound. The methods section contains all the necessary details, although the authors may also consider providing the name of the statistical analysis software in section 4.2.

The manuscript is fairly well written and easy to follow, although some linguistic errors and inconsistencies should be addressed. Examples are provided below:

Line 17: a predicate is missing in the sentence “Vitreous biopsies more likely to be culture-positive (41.1%) than anterior chamber biopsies (21.6%, p<0.0001).” The sentence should read “Vitreous biopsies are more likely to be culture-positive (41.1%) than anterior chamber biopsies (21.6%, p<0.0001).”

Line 30: the modifier “sight threatening” should use a hyphen: “sight-threatening disease”.

Line 69: the modifier “culture positive” should use a hyphen: “culture-positive biopsies”.

Line 33: a spelling error and incorrect usage of the articles: “United states” should be changed to “the United States”.

Line 89: the abbreviation SVL should be expanded at first mention in the text.

Figure 1 and 2: inconsistency in axis-x label: Time, months vs Time, m

Table 1 headings: Post cataract, post filtration etc. should be spelled either with a hyphen or no space.

Please note that these are examples only. I recommend a careful review of the manuscript to ensure that it is error free.

Author Response

Thank you for the review and comments of our manuscript (#antibiotics-1720081). We have addressed the reviewers’ comments individually and listed specific changes made to the manuscript in the form of a table as suggested.

Reviewer Comment

Response

Change

Reviewer 3

The study is well designed and methodologically sound. The methods section contains all the necessary details, although the authors may also consider providing the name of the statistical analysis software in section 4.2.

Added

Page 20 Line 8, " Analyses were performed by using SPSS statistical software version 22 (IBM, Chicago, IL)."

The manuscript is fairly well written and easy to follow, although some linguistic errors and inconsistencies should be addressed. Examples are provided below:

Line 17: a predicate is missing in the sentence “Vitreous biopsies more likely to be culture-positive (41.1%) than anterior chamber biopsies (21.6%, p<0.0001).” The sentence should read “Vitreous biopsies are more likely to be culture-positive (41.1%) than anterior chamber biopsies (21.6%, p<0.0001).”

Corrected

Page 2 Line 7, " Results: Vitreous biopsies were more likely to be culture-positive (41.1%) than anterior chamber biopsies (21.6%, p<0.0001)."

Line 30: the modifier “sight threatening” should use a hyphen: “sight-threatening disease”.

Corrected

Page 4 Line 2, "Endophthalmitis is a sight-threatening disease and has a high risk of poor visual outcome and even complete vision loss."

Line 69: the modifier “culture positive” should use a hyphen: “culture-positive biopsies”.

Corrected

Page 6 Line 15, "Antibiotic resistance patterns were tested for all culture-positive biopsies,"

Page 10 Line 7, " Visual outcome at 1 month was better for eyes with culture-negative than culture-positive biopsies"

Page 10 Line 8, " This difference remained significant throughout follow-up and by 12 months culture-negative eyes had a BCVA of 1.02±0.11 LogMAR compared with 1.74±0.14 LogMAR for culture-positive eyes (p<0.0001)."

Table 1

Line 33: a spelling error and incorrect usage of the articles: “United states” should be changed to “the United States”.

Corrected

Page 4 Line 5, " ranging from 0.13% to 0.7% in 1990s1, 2 to 0.04% in 2013 – 2017 in the United states."

Line 89: the abbreviation SVL should be expanded at first mention in the text.

Corrected

Page 8 Line 1, " At presentation 91.53% of eyes had vision loss (BCVA <20/40), of which 79.64% had Severe vision loss (SVL, ≤20/200)."

Figure 1 and 2: inconsistency in axis-x label: Time, months vs Time, m

Figure 1 changed

Figure 1

Table 1 headings: Post cataract, post filtration etc. should be spelled either with a hyphen or no space.

Corrected

Table 1

Please note that these are examples only. I recommend a careful review of the manuscript to ensure that it is error free.

We thank the reviewer for his assistance with linguistic editing. We made all the corrections as well as additional changes where found.

We corrected any grammar or spelling mistakes that were found.

We hope this revised manuscript is acceptable for publication in your journal.

Sincerely,

Oren Tomkins-Netzer MD, PhD

Reviewer 4 Report

Acute postoperative endophthalmitis (APE) is a common complication after ocular surgery, which is caused by the perioperative introduction of bacteria into the eye from the patient's conjunctival and skin flora or from external sources. The consequences of APE are associated with moderate to severe eye pain and vision loss. Regardless of emerging antibiotic resistance, the systemic or local administration of antibiotics is largely sufficient for clinical APE management. In this manuscript, the study includes 233 eyes with APE diagnosis. The authors performed the statistical analysis to investigate the antibiotic resistance and visual outcome. The results suggest that laboratory antibiotic resistance is not related to poor visual function. Overall, the study is well designed. I have a few suggestions:

  1. Please provide the statistical method in each figure legends

  1. Please provide the sample number (n) for each time point in Figure 1

  1. Some of the result interpretation is confusing, For example,

1) Ocular samples were culture-positive in 99 cases (44.8%), which should be (99/226=43.8%? );

2) Antibiotic resistance patterns were tested for all culture positive biopsies, and resistance included amikacin in 19 eyes (26.4%), vancomycin in 25 eyes (30.1%), and moxifloxacin in 13 eyes (16.7%). How those are calculated, please provide details.

Author Response

Thank you for the review and comments of our manuscript (#antibiotics-1720081). We have addressed the reviewers’ comments individually and listed specific changes made to the manuscript in the form of a table as suggested.

Reviewer Comment

Response

Change

Reviewer 4

Please provide the statistical method in each figure legends

Added to each figure

Figure 1

Figure 2

Figure 3

Please provide the sample number (n) for each time point in Figure 1

Added

Figure 1

Some of the result interpretation is confusing, For example,

1) Ocular samples were culture-positive in 99 cases (44.8%), which should be (99/226=43.8%? );

Thank you for your comment. We were able to find information on ocular sampling for 221 eyes. We added a clarification to the results.

Page 6 Line 2, "Information on ocular sampling was collected for 221 eyes (97.8%). "

2) Antibiotic resistance patterns were tested for all culture positive biopsies, and resistance included amikacin in 19 eyes (26.4%), vancomycin in 25 eyes (30.1%), and moxifloxacin in 13 eyes (16.7%). How those are calculated, please provide details.

We thank the reviewer for this comment. Although all positive samples were tested for sensitivity were able to find results for 77 eyes. We corrected the results accordingly.

Page 2 Line 7, " Results: Vitreous biopsies were more likely to be culture-positive (41.1%) than anterior chamber biopsies (21.6%, p<0.0001). Antibiotic resistance for amikacin was found in 19 eyes (24.7%), vancomycin in 29 eyes (31.5%) and moxiflocacin in 14 eyes (16.1%)."

Page 6 Line 15, "Antibiotic resistance patterns were tested for all culture-positive biopsies and information was available for 77 eyes (77.78%). Resistance included amikacin in 19 eyes (24.7%), vancomycin in 29 eyes (31.5%) and moxiflocacin in 14 eyes (16.1%). Among gram-positive bacteria, vancomycin resistance was found in five cases (7.8%). "

We corrected any grammar or spelling mistakes that were found.

We hope this revised manuscript is acceptable for publication in your journal.

Sincerely,

Oren Tomkins-Netzer MD, PhD

Round 2

Reviewer 2 Report

The article has been improved a lot, can be published.